# MULTI-SOURCE UNSUPERVISED HYPERPARAMETER OPTIMIZATION

## ABSTRACT

*How can we conduct efficient hyperparameter optimization for a completely new task?* In this work, we consider a novel setting, where we search for the optimal hyperparameters for a target task of interest using only unlabeled target task and 'somewhat relevant' source task datasets. In this setting, it is essential to estimate the ground-truth target task objective using only the available information. We propose estimators to unbiasedly approximate the ground-truth with a desirable variance property. Building on these estimators, we provide a general and tractable hyperparameter optimization procedure for our setting. The experimental evaluations demonstrate that the proposed framework broadens the applications of automated hyperparameter optimization.

## 1 INTRODUCTION

*Hyperparameter optimization* (HPO) has been a pivotal part of machine learning (ML) and contributed to achieving a good performance in a wide range of tasks (Feurer & Hutter, 2019). It is widely acknowledged that the performance of deep neural networks depends greatly on the configuration of the hyperparameters (Dacrema et al., 2019; Henderson et al., 2018; Lucic et al., 2018). HPO is formulated as a special case of a black-box function optimization problem, where the input is a set of hyperparameters, and the output is a validation score. Among the black-box optimization methods, adaptive algorithms, such as *Bayesian optimization* (BO) (Brochu et al., 2010; Shahriari et al., 2015; Frazier, 2018) have shown superior empirical performance compared with traditional algorithms, such as grid search or random search (Bergstra & Bengio, 2012).

One critical assumption in HPO is the **availability of an accurate validation score**. However, in reality, there are many cases where we cannot access the ground-truth of the task of interest (referred to as target task hereinafter). For example, in display advertising, predicting the effectiveness of each advertisement, i.e., *click-through rates* (CTR), is important for showing relevant advertisements (ads) to users. Therefore, it is necessary to conduct HPO before a new ad campaign starts. However, for new ads that have not yet been displayed to users, one cannot use labeled data to conduct HPO. In this case, the standard HPO procedure is infeasible, as one cannot utilize the labeled target task data and the true validation score of the ML model under consideration.

In this work, we address the infeasibility issue of HPO when the labels of the target task are unavailable. To formulate this situation, we introduce a novel HPO setting called *multi-source unsupervised hyperparameter optimization* (MSU-HPO). In MSU-HPO, it is assumed that we do not have the labeled data for a target task. However, we do have the data for some source tasks with a different distribution from the target task. It is natural to assume that we have access to multiple source tasks in most practical settings. In the display advertising example, several labeled datasets of old ads that have already been deployed are often available, which we can use as labeled source task datasets. To the best of our knowledge, no HPO approach that can address a situation without labeled target task data exists despite its significance and possibility for applications.

A problem with MSU-HPO is that the ground-truth is inaccessible, and one cannot directly apply the standard HPO procedure. Thus, it is essential to accurately approximate it using only available data. For this purpose, we propose two estimators, enabling the evaluation of the ML models without the labeled target task data. Our estimators are general and can be used in combination with any common black-box optimization methods, such Gaussian process-based BO (Srinivas et al., 2010; Snoek et al., 2012; Hennig & Schuler, 2012; Contal et al., 2014; Hernández-Lobato et al., 2014;

Wang & Jegelka, 2017) and the tree-structured Parzen estimator (Bergstra et al., 2011; 2013). In addition, we show that the proposed estimators can unbiasedly approximate the target task objective, one of which achieves a desirable variance property by selecting useful source tasks based on a task divergence measure. We also present a general and computationally inexpensive HPO procedure for MSU-HPO building on our estimators. Finally, we demonstrate that our estimators work properly through numerical experiments with synthetic and real-world datasets.

**Related Work.** A typical HPO setting is to find a better set of hyperparameters using a labeled target task of interest. As faster convergence is an essential performance metric of the HPO methods, the research community is moving on to the *multi-source* or *transfer* settings for which there are some previously solved related source tasks. By combining the additional source task information and the labeled target task dataset, it has been shown that one can improve the hyperparameter search efficiency, and thus reach a better solution with fewer evaluations (Bonilla et al., 2008; Bardenet et al., 2013; Swersky et al., 2013; Yogatama & Mann, 2014; Ramachandran et al., 2018; Springenberg et al., 2016; Poloczek et al., 2017; Wistuba et al., 2018; Feurer et al., 2018; Perrone et al., 2018; 2019; Salinas et al., 2019). A critical difference between the multi-source HPOs and our MSU-HPO settings is the **existence of labels for the target task**. Previous studies usually assume that analysts can utilize labeled target data. However, as discussed above, this is often unavailable, and thus, most of these methods are infeasible.

One possible solution to address the unavailablity of labeled target data is to use warm starting methods (Vanschoren, 2019), which aims to find good initial hyperparameters for the target task. *Learning Initialization* (LI) finds promising hyperparameters by minimizing a sum of a loss function surrogated by a Gaussian process on each source task (Wistuba et al., 2015). While LI is effective when the source and target tasks are quite similar, it is hard to achieve a reasonable performance otherwise. In contrast, *DistBO* learns the similarity between the source and target tasks with a joint Gaussian process model on hyperparameters and data representations (Law et al., 2019). However, many transfer methods including DistBO need abundant hyperparameter evaluations for the source tasks to surrogate objective function for each task well, which will be confirmed in our experiments.

Another related field is *model evaluation in covariate shift*, whose objective is to evaluate the performance of the ML models of the target task using only a relevant **single** source dataset (Sugiyama et al., 2007; You et al., 2019; Zhong et al., 2010). These studies build on the *importance sampling* (IS) method (Elvira et al., 2015; Sugiyama et al., 2007) to obtain an unbiased estimate of ground-truth model performances. While our proposed methods are also based on IS, a major difference is that we assume that there are multiple source datasets with different distributions. We demonstrate that with the multi-source setting, the previous IS method can fail, and propose an estimator satisfying the optimal variance property. Moreover, as these methods are specific to *model evaluation*, the connection between the IS-based estimation techniques and the automated HPO methods has not yet been explored despite their possible, broad applications. Consequently, we are the first to empirically evaluate the possible combination of the IS-based unbiased estimation and adaptive HPO.

**Contributions.** The contributions of this work can be summarized as follows: **(i):** We formulate a novel and highly practical HPO setting, MSU-HPO. **(ii):** We propose two unbiased estimators for the ground-truth validation score calculable with the available data. Additionally, we demonstrate that one of them achieves optimal finite variance among a reasonable class of unbiased estimators. **(iii):** We describe a flexible and computationally tractable HPO procedure building on the proposed estimators. **(iv):** We empirically demonstrate that the proposed procedure works favorably in MSU-HPO setting. Furthermore, our empirical results suggest a new possible connection between the adaptive HPO and IS-based unbiased estimation techniques.

## 2 PROBLEM SETTING

In this section, we formulate MSU-HPO. Let $\mathcal{X} \subseteq \mathbb{R}^d$ be the $d$-dimensional input space and $\mathcal{Y} \subseteq \mathbb{R}$ be the real-valued output space. We use $p_T(x, y)$ to denote the joint probability density function of the input and output variables $X \in \mathcal{X}$ and $Y \in \mathcal{Y}$ of the target task. The objective of this work is to find the best set of hyperparameters $\theta$ with respect to the target distribution:

$$\theta^{opt} = \arg\min_{\theta \in \Theta} f_T(\theta) \tag{1}$$

where $\Theta$ is a pre-defined hyperparameter search space and $f_T(\theta)$ is the target task objective, which is defined as the generalization error over the target distribution:

$$f_T(\theta) = \mathbb{E}_{(X,Y) \sim \mathcal{P}_T} [L(h_\theta(X), Y)] \tag{2}$$

where $L : \mathcal{Y} \times \mathcal{Y} \to \mathbb{R}_{\geq 0}$ is a bounded loss function such as the zero-one loss. $h_\theta : \mathcal{X} \to \mathcal{Y}$ is an arbitrary machine learning model that predicts the output values using the input vectors with a set of hyperparameters $\theta \in \Theta$.

In a standard hyperparameter optimization setting (Bergstra et al., 2011; Feurer & Hutter, 2019; Snoek et al., 2012), labeled i.i.d. validation samples $\{x_i, y_i\}_{i=1}^{n_T} \sim p_T$ are available, and one can easily estimate the target objective in Eq. (2) by the following empirical mean:

$$\hat{f}_T(\theta; \mathcal{D}_T^{labeled}) = \frac{1}{n_T} \sum_{i=1}^{n_T} L(h_\theta(x_i), y_i) \tag{3}$$

where $\mathcal{D}_T^{labeled}$ is any size $n_T$ of the i.i.d. labeled samples from the target task distribution. Then, a hyperparameter optimization is conducted directly using the estimated target function in Eq. (3) as a reasonable replacement for the ground-truth target objective $f_T(\theta)$ in Eq. (2).

In contrast, under the MSU-HPO setting, labels of the target task are assumed to be unobservable; we can use only **unlabeled** target validation samples denoted as $\mathcal{D}_T = \{x_i\}_{i=1}^{n_T}$ hereinafter. Instead, we assume the availability of the multiple *source task* datasets which is denoted as $\{\mathcal{D}_{S^j}\}_{j=1}^{N_S}$ where $j$ is a source task index and $N_S$ denotes the number of source tasks. Each source task data is defined as the i.i.d. **labeled** samples: $\mathcal{D}_{S^j} = \{x_i^j, y_i^j\}_{i=1}^{n_{Sj}} \sim p_{S^j}$ where $p_{S^j}(x, y)$ is a joint probability density function that characterizes the source task $j$. Note here that marginal input distributions of the target and source tasks are different, i.e., $p_T(x) \neq p_{S^j}(x), \ \forall j \in \{1, \ldots, N_S\}$.

Regarding the target and source distributions, we make the following assumptions.

**Assumption 1.** *Source tasks have support for the target task, i.e., $p_T(x) > 0 \Rightarrow p_{S^j}(x) > 0, \ \forall x \in \mathcal{X}, \ \forall j \in \{1, \ldots, N_S\}$.*

**Assumption 2.** *Conditional output distributions remain the same between the target and all of the source tasks, i.e., $p_T(y|x) = p_{S^j}(y|x), \ \forall j \in \{1, \ldots, N_S\}$.*

The above assumptions are common in the *covariate shift* literature Shimodaira (2000) and suggest that the input-output relation is the same, but the input distributions are different for the target and source task distributions. [1]

One critical difficulty of the MSU-HPO setting is that the simple approximation using the empirical mean is infeasible, as the labeled target dataset is unavailable. It is thus essential to accurately estimate the target task objective using only an unlabeled target dataset and labeled multiple source datasets.

## 3 METHOD

In this section, we propose estimators to approximate the target task objective by applying an importance weighting technique.

### 3.1 UNBIASED OBJECTIVE ESTIMATOR

A natural first candidate method to approximate the target task objective function is to use *importance weighting* (Shimodaira, 2000). To define our estimator, we first introduce the density ratio between the target task distribution and the source task distribution below.

**Definiton 1.** *(Density Ratio) For any $(x, y) \in \mathcal{X} \times \mathcal{Y}$ with a positive source density $p_{S^j}(x, y) > 0$, the density ratio between the target and a source task distributions is*

$$0 \leq w_{S^j}(x, y) = \frac{p_T(x, y)}{p_{S^j}(x, y)} = \frac{p_T(x)}{p_{S^j}(x)} = w_{S^j}(x) \leq C \tag{4}$$

*where $C$ is a positive constant. The equalities are derived from Assumption 2.*

---

[1] These assumptions seem to be strict, but in fact, they are relatively reasonable given that the general HPO literature implicitly assume that the train-test distributions are the same.

Using the density ratio, we define an estimator for the target task objective function.

**Definiton 2.** *(Unbiased Estimator) For a given set of hyperparameter $\theta \in \Theta$, the unbiased estimator for the target task objective function is defined as*

$$\hat{f}_{UB}\left(\theta; \{\mathcal{D}_{S^j}\}_{j=1}^{N_S}\right) = \frac{1}{n} \sum_{j=1}^{N_S} \sum_{i=1}^{n_{Sj}} w_{S^j}(x_i^j) \cdot L(h_\theta(x_i^j), y_i^j) \tag{5}$$

*where UB stands for unbiased, $n = \sum_{j=1}^{N_S} n_{S^j}$ is the total sample size of the source tasks, $\mathcal{D}_{S^j}$ is any sample size $n_{S^j}$ of the i.i.d. samples from the distribution of source task $j$, and $w_{S^j}(\cdot)$ is the true density ratio function.*

The estimator in Eq. (5) is an application of the *importance weighted cross-validation (Sugiyama et al., 2007)* to the multiple-source task setting and can easily be shown to be statistically unbiased for the ground-truth target task objective function, i.e., for any given $\theta$, $\mathbb{E}[\hat{f}_{UB}\left(\theta; \{\mathcal{D}_{S^j}\}_{j=1}^{N_S}\right)] = f_T(\theta)$.

We also characterize the variance of the unbiased estimator.

$$\mathbb{V}\left(\hat{f}_{UB}\left(\theta; \{\mathcal{D}_{S^j}\}_{j=1}^{N_S}\right)\right) = \frac{1}{n^2} \sum_{j=1}^{N_S} n_{S^j} \left(\mathbb{E}_{(X,Y)\sim p_{Sj}}\left[w_{S^j}^2(X) \cdot L^2(h_\theta(X), Y)\right] - (f_T(\theta))^2\right)$$

$$\tag{6}$$

As stated above, the unbiased estimator is a valid approach for approximating a target task objective because of its unbiasedness. The problem is that its variance depends on the square value of the density ratio function, which can be huge when there is a source task with a distribution that is dissimilar to that of the target task.

To illustrate this variance problem, we use a toy example where $\{x_1, x_2\} \subseteq \mathcal{X}$, $\{y_1, y_2\} \subseteq \mathcal{Y}$, $p(y_1|x_1) = p(y_2|x_2) = 1$, $p(y_2|x_1) = p(y_1|x_2) = 0$. The loss values for possible tuples, and the probability densities of the target and two source tasks are presented in Table 1. It shows that the target task $T$ is similar to the source task $S^2$, but its distribution is significantly different from that of $S^1$. For simplicity and without loss of generality, suppose there are two source task datasets such as $\mathcal{D}^1 = \{(x_1^1, y_1^1)\}$ and $\mathcal{D}^2 = \{(x_1^2, y_1^2)\}$. Then from Eq. (6), the variance of the unbiased estimator is about $64.27$. Intuitively, this large variance is a result of the large variance samples from $S^1$. In fact, by dropping the samples of $S^1$ reduces the variance to $4.27$. From this example, we know that the unbiased estimator fails to make the most of the source tasks, and there is room to improve its variance by down-weighting the source tasks dissimilar to the target task.

Table 1: Dropping data samples from $S^1$ significantly lowers the variance of the unbiased estimator

|  | $(x_1, y_1)$ | $(x_2, y_2)$ |
| --- | --- | --- |
| loss function: $L(h_\theta(x), y)$ | 10 | 1 |
| target task $(T)$ distribution: $p_T(x, y)$ | 0.8 | 0.2 |
| source task $(S^1)$ distribution: $p_{S^1}(x, y)$ | 0.2 | 0.8 |
| source task $(S^2)$ distribution: $p_{S^2}(x, y)$ | 0.9 | 0.1 |

## 3.2 Variance Reduced Objective Estimator

As illustrated with the toy example, an unbiased estimator can be unstable when there are some source tasks with a distribution significantly different from that of the target task. To address this variance issue, we define a *divergence measure* between the two tasks below.

**Definiton 3.** *(Task Divergence Measure) The divergence between a source task distribution $p_{S^j}$ where $j \in \{1, \ldots N_S\}$ and the target task distribution $p_T$ is defined as*

$$Div\left(T \mid\mid S^j\right) = \mathbb{E}_{(X,Y)\sim p_{Sj}}\left[w_{S^j}^2(X) \cdot L^2(h_\theta(X), Y)\right] - (f_T(\theta))^2 \tag{7}$$

This task divergence measure is large when the corresponding source distribution deviates significantly from the target task distribution. Building on this measure, we define the following estimator for the target task objective.

---

**Algorithm 1** Hyperparameter optimization procedure under the MSU-HPO setting

---

**Input:** unlabeled target task dataset $\mathcal{D}_T = \{x_i\}_{i=1}^{n_T}$; labeled source task datasets $\{\mathcal{D}_{Sj} = \{x_i^j, y_i^j\}_{i=1}^{n_{Sj}}\}_{j=1}^{N_S}$; hyperparameter search space $\Theta$; a machine learning model $h_\theta$; a target task objective estimator $\hat{f}$, a hyperparameter optimization algorithm **OPT**

1: **for** $j \in \{1, \ldots, N_S\}$ **do**
2:     Split $\mathcal{D}_{Sj}$ into three folds $\mathcal{D}_{Sj}^{density}$, $\mathcal{D}_{Sj}^{train}$, and $\mathcal{D}_{Sj}^{val}$
3:     Estimate density ratio $w_{Sj}(\cdot)$ by uLSIF with $\mathcal{D}_T$ and $\mathcal{D}_{Sj}^{density}$
4: **end for**
5: Optimize the hyperparameter $\theta \in \Theta$ of $h_\theta$ with **OPT** by setting $\hat{f}(\theta; \{\mathcal{D}_{Sj}^{val}\}_{j=1}^{N_S})$ as its objective
6:     (the model parameter of $h_\theta$ is obtained by optimizing $\hat{f}(\theta; \{\mathcal{D}_{Sj}^{train}\}_{j=1}^{N_S})$)
7: **return** $h_{\theta^\star}$ (where $\theta^\star$ is the output of **OPT**)

---

**Definiton 4.** *(Variance Reduced Estimator) For a given set of hyperparameters $\theta \in \Theta$, the variance reduced estimator for the target task objective function is defined as*

$$\hat{f}_{VR}\left(\theta; \{\mathcal{D}_{Sj}\}_{j=1}^{N_S}\right) = \sum_{j=1}^{N_S} \lambda_j^\star \sum_{i=1}^{n_{Sj}} w(x_i^j) \cdot L(h_\theta(x_i^j), y_i^j) \tag{8}$$

*where VR stands for variance reduced, $\mathcal{D}_{Sj}$ is any sample size $n_{Sj}$ of the i.i.d. samples from the distribution of source task $j$, and $w_{Sj}(\cdot)$ is the true density ratio function. $\lambda_j^\star$ is a weight for source task $j$, which is defined as $\lambda_j^\star = \left(Div\left(T \| S^j\right) \sum_{j=1}^{N_S} \frac{n_{Sj}}{Div(T \| S^j)}\right)^{-1}$. Note that, for all $j \in \{1, \ldots N_S\}$, $\lambda_j^\star \geq 0$ and $\sum_{j=1}^{N_S} \lambda_j^\star n_{Sj} = 1$.*

The variance reduced estimator in Eq. (8) is also statistically unbiased for the ground-truth target task objective in Eq. (2), i.e., for any given $\theta$, $\mathbb{E}[\hat{f}_{VR}\left(\theta; \{\mathcal{D}_{Sj}\}_{j=1}^{N_S}\right)] = f_T(\theta)$.

Then, we demonstrate that the variance reduced estimator in Eq. (8) is optimal in the sense that any other convex combination of a set of weights $\boldsymbol{\lambda} = \{\lambda_1, \ldots \lambda_{N_S}\}$ that satisfies the unbiasedness for the target task objective function does not provide a smaller variance.

**Theorem 1.** *(Variance Optimality; Extension of Theorem 6.4 of (Agarwal et al., 2017)) For any given set of weights $\boldsymbol{\lambda} = \{\lambda_1, \ldots \lambda_{N_S}\}$ that satisfies $\lambda_j \geq 0$ and $\sum_{j=1}^{N_S} \lambda_j n_{Sj} = 1$ for all $j \in \{1, \ldots N_S\}$, the following inequality holds*

$$\mathbb{V}\left(\hat{f}_{VR}\left(\theta; \{\mathcal{D}_{Sj}\}_{j=1}^{N_S}\right)\right) = \left(\sum_{j=1}^{N_S} \frac{n_{Sj}}{Div\left(T \| S^j\right)}\right)^{-1} \leq \mathbb{V}\left(\hat{f}_{\boldsymbol{\lambda}}\left(\theta; \{\mathcal{D}_{Sj}\}_{j=1}^{N_S}\right)\right)$$

*where $\hat{f}_{\boldsymbol{\lambda}}(\theta; \{\mathcal{D}_{Sj}\}_{j=1}^{N_S}) = \sum_{j=1}^{N_S} \lambda_j \sum_{i=1}^{n_{Sj}} w(x_i^j) \cdot L(h_\theta(x_i^j), y_i^j)$. See Appendix A for the proof.*

Theorem 1 suggests that the variance reduced estimator achieves a desirable finite sample variance property by weighting each source task based on its divergence to the target task.

Let us now return to the toy example in Table 1. The values of the divergence measure for $S^1$ and $S^2$ are 252.81 and 4.27, respectively. This leads to the weights of $\lambda_1^\star \approx 0.017$ and $\lambda_2^\star \approx 0.983$. Then, the variance of the variance reduced estimator is equal to $4.21 < 4.27$ (variance when $S^1$ is dropped.). It is obvious that the variance reduced estimator performs better than the unbiased estimator does by optimally weighting all available source tasks.

### 3.3 HYPERPARAMETER OPTIMIZATION PROCEDURE

We describe several detailed components of the HPO procedure in the MSU-HPO setting.

**Density Ratio Estimation:**     In general, density ratio functions between the target and source tasks are unavailable and thus should be estimated beforehand. To estimate this parameter, we employ the

*unconstrained Least-Squares Importance Fitting (uLSIF)* procedure Kanamori et al. (2009); Yamada et al. (2011), which suggests directly minimizing the following squared error for the true density ratio function:

$$\hat{s} = \arg\min_{s \in \mathcal{S}} \mathbb{E}_{p_{Sj}} \left[ (w(X) - s(X))^2 \right] = \arg\min_{s \in \mathcal{S}} \left[ \frac{1}{2} \mathbb{E}_{p_{Sj}} \left[ s^2(X) \right] - \mathbb{E}_{p_T} [s(X)] \right] \quad (9)$$

where $\mathcal{S}$ is a class of measurable functions. It should be noted that the empirical version of Eq. (9) is calculable with unlabeled target and source task datasets.

**Task Divergence Estimation:** To utilize the variance reduced estimator, the task divergence measure $Div\left(T \,\|\, S^j\right)$ in Eq. (7) needs to be estimated from the available data. This can be done using the following empirical mean. [2]

**How to train $h_\theta$?:** To evaluate the validation score of $\theta \in \Theta$, the model parameters of $h_\theta$ should be optimized by the supervised learning procedure. However, in the MSU-HPO setting, the labeled target task dataset is unavailable, and direct training of $h_\theta$ is infeasible. Therefore, we suggest splitting the labeled source task datasets $\{\mathcal{D}_{Sj}\}$ into the training $\{\mathcal{D}_{Sj}^{train}\}$ and validation $\{\mathcal{D}_{Sj}^{val}\}$ sets. Then, we can train $h_\theta$ using the training set by $h_\theta^* = \arg\min_{h_\theta \in \mathcal{H}_\theta} \hat{f}(\theta; \{\mathcal{D}_{Sj}^{train}\}_{j=1}^{N_S})$. where $\hat{f}$ is an estimator for the target task objective function such as the unbiased and variance reduced estimators, and $\mathcal{H}_\theta$ is a hypothesis space defined by a set of hyperparameters $\theta \in \Theta$. This training procedure enables us to obtain the model parameters of $h_\theta$ as if it were trained on the labeled target task dataset. In addition, it is sufficient to train $h_\theta$ only once to evaluate $\theta \in \Theta$; the proposed procedure is computationally inexpensive.

**Overall Procedure:** Building on the above details, Algorithm 1 summarizes the high-level hyperparameter optimization procedure under the MSU-HPO setting[3]. We also provide the regret bound of our HPO procedure in the MSU-HPO setting in Appendix C.

## 4 Experiments

We investigate the behavior of our proposed HPO procedure in MSU-HPO using a synthetic problem in Section 4.1 and real-world datasets in Section 4.2. We compare the following methods as possible baselines [4]: **(i)** *Learning Initialization (LI)* (Wistuba et al., 2015), **(ii)** *DistBO* (Law et al., 2019), **(iii):** *Naive* method, which uses the performance on the concatenation of source tasks as a validation score, **(iv)** *Oracle* method, which uses the labeled target task for HPO. Thus, the oracle method is infeasible in MSU-HPO, and we regard the performance of the oracle method as an upper bound to which other methods can reach. In all the experiments, we set the number of evaluations $B = 50$ and use GP-UCB (Srinivas et al., 2010) as a hyperparameter optimization algorithm[5].

### 4.1 Toy Problem

We consider a 1-dimensional regression problem with the MSU-HPO setting. The generative process of the toy dataset is as follows:

$$\mu^i \sim \mathcal{U}(-c_i, c_i), \ \{x_l^i\}_{l=1}^n \mid \mu^i \overset{i.i.d.}{\sim} \mathcal{N}(\mu^i, 1), \ \{y_l^i\}_{l=1}^n \mid \{x_l^i\}_{l=1}^n \overset{i.i.d.}{\sim} \{\mathcal{N}(0.7x_l^i + 0.3, 1)\}_{l=1}^n,$$

where $\mathcal{U}$ is the uniform distribution, $\mathcal{N}$ denotes the normal distribution, and $c_i \in \mathbb{R}$ is a prior parameter that characterizes the marginal input distribution ($p(x)$) of task $i$. The objective function $f$ is given by:

$$f(\theta; \mathcal{D}_i) = \frac{1}{n} \sum_{l=1}^n L(\theta, y_l), \ L(\theta, y_l) = (\theta - y_l)^2 / 2. \quad (10)$$

---

[2] $\widehat{Div}\left(T \,\|\, S^j\right) = \frac{1}{n_{Sj}} \sum_{i=1}^{n_{Sj}} \left( w(x_i^j) \cdot L(h_\theta(x_i^j), y_i^j) \right)^2 - \left( \frac{1}{n_{Sj}} \sum_{i=1}^{n_{Sj}} w(x_i^j) \cdot L(h_\theta(x_i^j), y_i^j) \right)^2$

[3] We describe the specific hyperparameter optimization procedure when BO is used as **OPT** in Appendix B.

[4] We describe the details of the baseline methods in Appendix D.

[5] We describe detailed settings in our experiments in Appendix E.

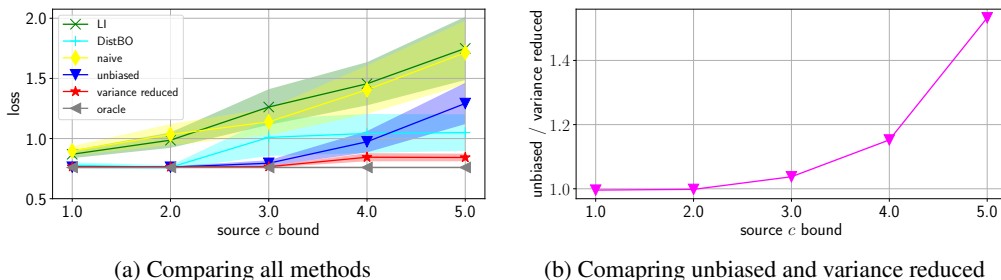

(a) Comparing all methods         (b) Comapring unbiased and variance reduced

Figure 1: Results of the experiment on synthetic toy problems over 30 runs.

*Note*: The horizontal axis represents the prior parameters of the source tasks $c_i^S \in \{1.0, \cdots, 5.0\}$, $(i \in \{1, \cdots, N^S\})$. (a) The vertical axis represents the mean and standard error of the evaluated loss for each estimator. (b) The vertical axis represents the ratio of the mean loss of the unbiased estimator to the variance reduced estimator.

Similar to the toy experiment in (Law et al., 2019), $\theta \in [-8, 8]$ is a hypothetical 'hyperparameter' we would like to optimize. The optimal solution for this experiment is thus $\theta = n^{-1} \sum_{l=1}^{n} y_l$.

We described in Section 3 that when $p(x)$ of the source task and the target task differs significantly, the performance of the variance reduced estimator is better than that of the unbiased estimator. To demonstrate this, we set $c_i$ separately for the source ($c_i^S \in \{1.0, 2.0, \cdots, 5.0\}, i \in \{1, \cdots, N_S\}$) and the target tasks ($c^T = 1.0$). That is, the source and target distributions are similar when $c_i^S = 1.0 (= c^T)$; in contrast, the source and target distributions are quite different when $c_i^S = 5.0$. Finally, we set $N^S = 2$ and $n = 1000$.

Figure 1 shows the results of the experiment on the toy problem over 30 runs with different random seeds. First, Figure 1 (a) indicates that the proposed unbiased and variance reduced estimators significantly outperform the naive method and LI in all settings. This is because our estimators can unbiasedly approximate the target task objective by considering the distributional shift, while the naive method and LI cannot. Moreover, this figure shows the advantage of unbiasedness is highlighted when the distributions of the target and source tasks diverge largely (i.e., when $c_i^S$ is large.). DistBO also shows relatively good performance despite the lack of unbiasedness. Next, we compare the performance of the unbiased and variance reduced estimator in Figure 1 (b). This reports the performance of the unbiased estimator relative to the variance reduced one with varying values of $c$. The result indicates that the advantages of using the variance reduced estimator over the unbiased one are further strengthened when there is a large divergence between the target and source task distributions, which is consistent with our theoretical analysis. Finally, as shown in Figure 1 (a), the variance reduced estimator achieves almost the same performance as the upper bound without using the labels of the target task, suggesting its powerful HPO performance on an unlabeled target task.

### 4.2 Hyperparameter Optimization on Real-World Datasets

**Datasets:** We use Parkinson's telemonitoring (Parkinson) (Tsanas et al., 2009) and Graft-versus-host disease (GvHD) datasets (Brinkman et al., 2007) to evaluate our methods on real-world problems.

Parkinson data consists of voice measurements of 42 patients with the early-stage Parkinson disease collected by using a telemonitoring device in remote symptom progression monitoring. Each patient has about 150 recordings characterized by a feature vector with 17 dimensions. The goal is to predict the Parkinson disease symptom score for each recording from the recordings.

GvHD is an important medical problem in the allogeneic blood transplantation field (Brinkman et al., 2007). The issue occurs in allogeneic hematopoietic stem cell transplant recipients when donor-immune cells in the graft recognize the recipient as foreign and initiate an attack on several tissues. The GvHD dataset contains weekly peripheral blood samples obtained from 31 patients characterized by a feature vector with 7 dimensions. Following (Muandet et al., 2013), we omit one patient who has insufficient data, and subsample data of each patient to have 1000 data points each.

The goal is to classify CD3+CD4+CD8+ cells, which have a high correlation with the development of the disease (Brinkman et al., 2007).

**Experimental Procedure:** To create the MSU-HPO setting, for both datasets, we treat each patient as a task. We select one patient as a target task and regard the remaining patients as multiple source tasks. Then, we use the following experimental procedure: (1) Tune hyperparameters of **an ML model** by an HPO method using the unlabeled target task and labeled source tasks, (2) Split the original target task data into 70% training set and 30% test set, (3) Train **an ML model** tuned by an MSU-HPO method using the training set of the target task, (4) Predict target variables (symptom scores for Parkinson and CD3+CD4+CD8+ cells for GvHD) on the test set of the target patient, (5) Calculate **target task objective** of the prediction and regard it as the performance of the MSU-HPO method under consideration, (6) Repeat the above steps 10 times with different seeds and report the mean and standard error over the simulations.

As for an ML model and a target task objective, we use *support vector machine* (SVM) implemented in *scikit-learn* (Pedregosa et al., 2011) and *mean absolute error* (MAE) for Parkinson. In contrast, we use LightGBM (Ke et al., 2017) as an ML model and *binary cross-entropy* (BCE) as a target task objective for GvHD.

Table 2: Comparing different MSU-HPO methods (Mean $\pm$StdErr)

| Estimators | Parkinson (MAE) | GvHD (BCE) |
|---|---|---|
| LI | 0.41507 $\pm$0.1669 | 0.19695 $\pm$0.0468 |
| DISTBO | 1.54202 $\pm$0.1006 | 0.33015 $\pm$0.0600 |
| NAIVE | 1.10334 $\pm$0.0908 | 0.02121 $\pm$0.0052 |
| UNBIASED (ours) | 1.08283 $\pm$0.1981 | 0.02141 $\pm$0.0052 |
| VARIANCE REDUCED (ours) | **0.40455** $\pm$**0.1755** | **0.01791** $\pm$**0.0039** |
| ORACLE (reference) | 0.06862 $\pm$0.0011 | 0.01584 $\pm$0.0043 |

*Note*: The red fonts represent the best performance among estimators using only the unlabeled target task and labeled source task datasets. The mean and standard error (StdErr) are induced by running 10 simulations with different random seeds.

**Results:** Table 2 presents the results of the experiments over 10 runs with different random seeds. In contrast to the results in Section 4.1, the performance of DistBO has deteriorated significantly. While DistBO requires a reasonable number of hyperparameter evaluations per source task, our setting allows only a very small number of evaluations per source task, which may lead to learning inaccurate surrogate models[6]. The unbiased estimator performs almost the same with naive in Parkinson given their standard errors. Moreover, it slightly underperforms the naive in GvHD, although the unbiased estimator satisfies the unbiasedness. This is because the number of data for each task is small, and the variance issue of the unbiased estimator is highlighted in these data. Therefore, pursuing only unbiasedness in the approximation of the target task objective is not sufficient in MSU-HPO. On the other hand, the variance reduced estimator alleviates the instability issue of the unbiased estimator and performs best in both datasets. The results also suggest that the variance reduced estimator works well on both regression (Parkinson) and classification (GvHD) tasks. Therefore, we conclude from its variance optimality and empirical performance that using the variance reduced estimator is the best choice for MSU-HPO.

## 5 CONCLUSION

We studied a novel problem setting, MSU-HPO, with the goal of enabling effective HPO with only an unlabeled target task and multiple labeled source task datasets. We proposed two estimators to approximate the target task objective from available data. Empirical evaluations demonstrated that the proposed HPO procedure helps to determine useful hyperparameters without the labels of the target task.

---

[6]A more detailed discussion is provided in Appendix D.3.

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

## A  OMITED PROOFS

### A.1  DERIVATION OF UNBIASEDNESS

We first define a general class of unbiased estimators called $\boldsymbol{\lambda}$-*unbiased estimator* that includes the unbiased and variance reduced estimators as special cases.

***Definiton 5.*** *($\boldsymbol{\lambda}$-unbiased Estimator) When a set of weights $\boldsymbol{\lambda} = \{\lambda_1, \ldots \lambda_{N_S}\}$ that satisfies $\lambda_j \geq 0$ and $\sum_{j=1}^{N_S} \lambda_j n_{Sj} = 1$ for all $j \in \{1, \ldots N_S\}$ is given, the $\boldsymbol{\lambda}$-unbiased estimator for the target task objective function is*

$$\hat{f}_{\boldsymbol{\lambda}}\left(\theta; \{\mathcal{D}_{Sj}\}_{j=1}^{N_S}\right) = \sum_{j=1}^{N_S} \lambda_j \sum_{i=1}^{n_{Sj}} w_{Sj}(x_i^j) \cdot L(h_\theta(x_i^j), y_i^j). \tag{11}$$

*When $\lambda_j = n_{Sj}/N$, it is the unbiased estimator in Eq. (5). In contrast, it is the variance reduced estimator in Eq. (8) when $\lambda_j = \lambda_j^\star$*

Then we show that the $\boldsymbol{\lambda}$-unbiased estimator is statistically unbiased for the target task function.

*Proof.* By the linearity of the expectation operator,

$$\mathbb{E}\left[\hat{f}_{\boldsymbol{\lambda}}\left(\theta; \{\mathcal{D}_{Sj}\}_{j=1}^{N_S}\right)\right] = \sum_{j=1}^{N_S} \lambda_j \sum_{i=1}^{n_{Sj}} \mathbb{E}_{(X,Y)\sim p_{Sj}}\left[w_{Sj}(X) \cdot L(h_\theta(X), Y)\right]$$

$$= \sum_{j=1}^{N_S} \lambda_j \sum_{i=1}^{n_{Sj}} \mathbb{E}_{(X,Y)\sim p_{Sj}}\left[\frac{p_T(X,Y)}{p_{Sj}(X,Y)} \cdot L(h_\theta(X), Y)\right]$$

$$= \sum_{j=1}^{N_S} \lambda_j \sum_{i=1}^{n_{Sj}} \mathbb{E}_{(X,Y)\sim p_T}\left[L(h_\theta(X), Y)\right]$$

$$= \sum_{j=1}^{N_S} \lambda_j \sum_{i=1}^{n_{Sj}} f_T(\theta)$$

$$= \left(\sum_{j=1}^{N_S} \lambda_j n_{Sj}\right) \cdot f_T(\theta)$$

$$= f_T(\theta)$$

Thus, the unbiased estimator in Eq. (5) and the variance reduced estimator in Eq. (8) are both statistically unbiased for the ground truth target task objective function in Eq. (2).  □

### A.2  DERIVATION OF EQ. (6)

*Proof.* The variance can be represented as follows because samples are independent

$$\mathbb{V}\left(\hat{f}_{UB}\left(\theta; \{\mathcal{D}_{Sj}\}_{j=1}^{N_S}\right)\right) = \frac{1}{n^2} \sum_{j=1}^{N_S} \sum_{i=1}^{n_{Sj}} \mathbb{V}\left(w_{Sj}(X) \cdot L(h_\theta(X), Y)\right)$$

$$= \frac{1}{n^2} \sum_{j=1}^{N_S} n_{Sj} \cdot \mathbb{V}\left(w_{Sj}(X) \cdot L(h_\theta(X), Y)\right)$$

$\mathbb{V}\left(w_{Sj}(X) \cdot L(h_\theta(X), Y)\right)$ is decomposed as

$$\mathbb{V}\left(w_{Sj}(X) \cdot L(h_\theta(X), Y)\right) = \mathbb{E}_{(X,Y)\sim p_{Sj}}\left[w_{Sj}^2(X) \cdot L^2(h_\theta(X), Y)\right] - \left(\mathbb{E}_{(X,Y)\sim p_{Sj}}\left[w_{Sj}(X) \cdot L(h_\theta(X), Y)\right]\right)^2$$

From the unbiasedness property, $\mathbb{E}_{(X,Y)\sim p_{Sj}}\left[w_{Sj}(X) \cdot L(h_\theta(X), Y)\right] = f_T(\theta)$. Then, we now have

$$\mathbb{V}\left(w_{Sj}(X) \cdot L(h_\theta(X), Y)\right) = \mathbb{E}_{(X,Y)\sim p_{Sj}}\left[w_{Sj}^2(X) \cdot L^2(h_\theta(X), Y)\right] - (f_T(\theta))^2$$

□

A.3 PROOF OF THEOREM 1

By following the same logic flow as in Section A.2, the variance of the $\boldsymbol{\lambda}$-unbiased estimator in Eq. (11) is

$$\mathbb{V}\left(\hat{f}_{\boldsymbol{\lambda}}\left(\theta; \{\mathcal{D}_{S^j}\}_{j=1}^{N_S}\right)\right) = \sum_{j=1}^{N_S} \lambda_j^2 n_{S^j} \left(\mathbb{E}_{(X,Y)\sim p_{S^j}}\left[w_{S^j}^2(X) \cdot L^2(h_\theta(X), Y)\right] - (f_T(\theta))^2\right)$$

$$= \sum_{j=1}^{N_S} \lambda_j^2 n_{S^j} \cdot Div\left(T \,\|\, S^j\right) \tag{12}$$

Thus, by replacing $\lambda_j$ for $\left(Div\left(T \,\|\, S^j\right)\sum_{j=1}^{N_S} \frac{n_{S^j}}{Div(T \,\|\, S^j)}\right)^{-1}$, we have

$$\mathbb{V}\left(\hat{f}_{\boldsymbol{\lambda}}\left(\theta; \{\mathcal{D}_{S^j}\}_{j=1}^{N_S}\right)\right) = \sum_{j=1}^{N_S} \left(\sum_{j=1}^{N_S} \frac{n_{S^j}}{Div\left(T \,\|\, S^j\right)}\right)^{-2} n_{S^j} \cdot Div\left(T \,\|\, S^j\right)$$

$$= \sum_{j=1}^{N_S} \frac{n_{S^j} Div\left(T \,\|\, S^j\right)}{(Div\left(T \,\|\, S^j\right))^2 (\sum_{j=1}^{N_S} \frac{n_{S^j}}{Div(T \,\|\, S^j)})^2}$$

$$= \left(\sum_{j=1}^{N_S} \frac{n_{S^j}}{Div\left(T \,\|\, S^j\right)}\right)\left(\sum_{j=1}^{N_S} \frac{n_{S^j}}{Div\left(T \,\|\, S^j\right)}\right)^{-2}$$

$$= \left(\sum_{j=1}^{N_S} \frac{n_{S^j}}{Div\left(T \,\|\, S^j\right)}\right)^{-1}$$

Moreover, for any set of weights $\boldsymbol{\lambda} = \{\lambda_1, \ldots \lambda_{N_S}\}$, we obtain the following variance optimality using the Cauchy-Schwarz inequality.

$$\left(\sum_{j=1}^{N_S} \lambda_j^2 n_{S^j} \cdot Div\left(T \,\|\, S^j\right)\right)\left(\sum_{j=1}^{N_S} \frac{n_{S^j}}{Div\left(T \,\|\, S^j\right)}\right) \geq \left(\sum_{j=1}^{N_S} \lambda_j n_{S^j}\right)^2 = 1$$

$$\implies \left(\sum_{j=1}^{N_S} \lambda_j^2 n_{S^j} \cdot Div\left(T \,\|\, S^j\right)\right) \geq \left(\sum_{j=1}^{N_S} \frac{n_{S^j}}{Div\left(T \,\|\, S^j\right)}\right)^{-1}$$

$$\implies \mathbb{V}\left(\hat{f}_{\boldsymbol{\lambda}}\left(\theta; \{\mathcal{D}_{S^j}\}_{j=1}^{N_S}\right)\right) \geq \mathbb{V}\left(\hat{f}_{VR}\left(\theta; \{\mathcal{D}_{S^j}\}_{j=1}^{N_S}\right)\right)$$

## B  BAYESIAN OPTIMIZATION UNDER THE MSU-HPO SETTING

In Algorithm 1, we described the abstracted hyperparameter optimization procedure which allows any black-box optimization method to be used. Here, in Algorithm 2, we describe the hyperparameter optimization procedure under the MSU-HPO setting with the popular Bayesian optimization method.

## C  REGRET ANALYSIS

In this section, we analyze the regret bound under the MSU-HPO setting. We define a *regret* as

$$r_B^n = f(\hat{\theta}_B^*) - f(\theta^*),$$

where $f : \Theta \to \mathbb{R}$ is the ground-truth target task objective, $n = \sum_{j=1}^{N_S} n_{S^j}$ is the total sample size among source tasks, $B$ is the total number of evaluations, $\theta^* = \arg\min_{\theta \in \Theta} f(\theta)$, and $\hat{\theta}_B^* = \arg\min_{\theta \in \{\theta_1, \cdots, \theta_B\}} \hat{f}_n(\theta)$ where $\hat{f}_n : \Theta \to \mathbb{R}$ is a estimated target task objective by any estimator

---

**Algorithm 2** Bayesian Optimization under the MSU-HPO setting

---

**Input:** unlabeled target task dataset $\mathcal{D}_T = \{x_i\}_{i=1}^{n_T}$; labeled source task datasets $\{\mathcal{D}_{Sj} = \{x_i^j, y_i^j\}_{i=1}^{n_{Sj}}\}_{j=1}^{N_S}$; hyperparameter search space $\Theta$; a machine learning model $h_\theta$; a target task objective estiamtor $\hat{f}$, number of evaluations $B$, acquisition function $\alpha(\cdot)$

**Output:** the optimized set of hyperparameters $\theta^\star \in \Theta$

 1: Set $\mathcal{A}_0 \leftarrow \emptyset$
 2: **for** $j \in \{1, \ldots, N_S\}$ **do**
 3:     Split $\mathcal{D}_{Sj}$ into three folds $\mathcal{D}_{Sj}^{density}$, $\mathcal{D}_{Sj}^{train}$, and $\mathcal{D}_{Sj}^{val}$
 4:     Estimate density ratio $w_{Sj}(\cdot)$ by uLSIF with $\mathcal{D}_T$ and $\mathcal{D}_{Sj}^{density}$
 5: **end for**
 6: **for** $t = 1, 2, \ldots, B$ **do**
 7:     Select $\theta_t$ by optimizing $\alpha(\theta \mid \mathcal{A}_{t-1})$
 8:     Train $h_{\theta_t}$ by optimizing $\hat{f}(\theta; \{\mathcal{D}_{Sj}^{train}\}_{j=1}^{N_S})$ and obtain a trained model $h_\theta^*$
 9:     Evaluate $h_\theta^*$ and obtain a validation score $z_t = \hat{f}(\theta; \{\mathcal{D}_{Sj}^{val}\}_{j=1}^{N_S})$
10:     $\mathcal{A}_t \leftarrow \mathcal{A}_{t-1} \cup \{(\theta_t, z_t)\}$
11: **end for**
12: $t^\star = \arg\min_t\{z_1, \ldots z_B\}$
13: **return** $h_{\theta^\star}$ (where $\theta^\star = \theta_{t^\star}$)

---

(e.g., the unbiased estimator and the variance reduced estimator). Note that each of $\{\theta_1, \cdots, \theta_B\}$ is the hyperparameter selected in $B$ evaluations in the optimization.

To bound the regret above, we first decompose it into the following terms:

$$
\begin{aligned}
r_B^n &= f(\hat{\theta}_B^*) - f(\theta^*) \\
&= (f(\hat{\theta}_B^*) - \hat{f}_n(\hat{\theta}_B^*)) + \hat{f}_n(\hat{\theta}_B^*) + (\hat{f}_n(\hat{\theta}^*) - f(\theta^*)) - \hat{f}_n(\hat{\theta}^*) \\
&= \underbrace{(\hat{f}_n(\hat{\theta}_B^*) - \hat{f}_n(\hat{\theta}^*))}_{(A)} + \underbrace{(f(\hat{\theta}_B^*) - \hat{f}_n(\hat{\theta}_B^*))}_{(B)} + \underbrace{(\hat{f}_n(\hat{\theta}^*) - f(\theta^*))}_{(C)},
\end{aligned}
\tag{13}
$$

where $\hat{\theta}^* = \arg\min_{\theta \in \Theta} \hat{f}_n(\theta)$.

The term (A) represents the regret obtained by optimizing the estimated target task objective $\hat{f}_n$. The term (B) represents the difference of a function value between the true objective $f$ and the estimated objective $\hat{f}_n$ at $\hat{\theta}_B^*$, which is the solution obtained by the optimization for the estimated objective. The term (C) represents the difference between the minimum value for the estimated objective $\hat{f}_n$ and that of the true objective $f$.

We first show the following two lemmas which is used to bound the regret.

**Lemma 2.** *The following inequality holds with a probability of at least $1 - \delta$, $\delta \in (0, 1)$*

$$
(f(\hat{\theta}_B^*) - \hat{f}_n(\hat{\theta}_B^*)) \le \sqrt{\mathbb{V}(\hat{f}_n(\hat{\theta}_B^*))/\delta}.
$$

*Proof.* By Chebyshev's inequality, we have

$$
\begin{aligned}
\mathbb{P}\{f(\hat{\theta}_B^*) - \hat{f}_n(\hat{\theta}_B^*) \ge c\} &\le \mathbb{P}\{|f(\hat{\theta}_B^*) - \hat{f}_n(\hat{\theta}_B^*)| \ge c\} \\
&\le \mathbb{V}(\hat{f}_n(\hat{\theta}_B^*))/c^2.
\end{aligned}
$$

Putting the RHS as $\delta$ and solving it for $c$ completes the proof. □

**Lemma 3.** *The following inequality holds with a probability of at least $1 - \delta$, $\delta \in (0, 1)$*

$$
\hat{f}_n(\hat{\theta}^*) - f(\theta^*) \le \sqrt{(\mathbb{V}(\hat{f}_n(\theta^*)) + \mathbb{V}(\hat{f}_n(\hat{\theta}^*)))/\delta}.
$$

*Proof.* By Chebyshev's inequality, we have

$$\mathbb{P}\{\hat{f}_n(\hat{\theta}^*) - f(\theta^*) \geq c\}$$
$$\leq \mathbb{P}\{|\hat{f}_n(\hat{\theta}^*) - f(\theta^*)| \geq c\}$$
$$\leq \mathbb{P}\{|\hat{f}_n(\theta^*) - f(\theta^*)| \geq c \cup |\hat{f}_n(\hat{\theta}^*) - f(\hat{\theta}^*)| \geq c\}$$
$$\leq \mathbb{P}\{|\hat{f}_n(\theta^*) - f(\theta^*)| \geq c\} + \mathbb{P}\{|\hat{f}_n(\hat{\theta}^*) - f(\hat{\theta}^*)| \geq c\}$$
$$\leq \frac{1}{c^2}(\mathbb{V}(\hat{f}_n(\theta^*)) + \mathbb{V}(\hat{f}_n(\hat{\theta}^*))).$$

Putting the RHS as $\delta$ and solving it for $c$ completes the proof. ☐

**Theorem 4.** *(Regret Bound on the MSU-HPO setting) When the $\boldsymbol{\lambda}$-unbiased estimator with an arbitrary set of weights $\boldsymbol{\lambda}$ is used as $\hat{f}(\theta, ; \{\mathcal{D}_{S^j}^{val}\}_{j=1}^{N_S})$, the following regret bound holds with a probability of at least $1 - \delta$, $\delta \in (0, 1)$,*

$$r_B^n \leq R_n + \sqrt{2\mathbb{V}(\hat{f}_n(\hat{\theta}_B^*))/\delta} + \sqrt{2(\mathbb{V}(\hat{f}_n(\theta^*)) + \mathbb{V}(\hat{f}_n(\hat{\theta}^*)))/\delta}, \quad (14)$$

*where $R_n = \hat{f}_n(\hat{\theta}_B^*) - \hat{f}_n(\hat{\theta}^*)$.*

*Proof.* Putting Lemma 2 to the term (B) in Eq. (13) and Lemma 3 to the term (C) in Eq. (13) complete the proof. ☐

When an estimator is the proposed unbiased estimator or variance reduced esitmator, the variance $\mathbb{V}(\hat{f}_n(\cdot))$ is $o(n)$; the second term and third term in Eq. (14) is to be *n*o-regret (Srinivas et al., 2010) with respect to $n$, i.e., $\lim_{n \to \infty}(\sqrt{2\mathbb{V}(\hat{f}_n(\hat{\theta}_B^*))/\delta} + \sqrt{2(\mathbb{V}(\hat{f}_n(\theta^*)) + \mathbb{V}(\hat{f}_n(\hat{\theta}^*)))/\delta})/n = 0$. This means that, if the optimization method is no-regret with respect to the number of evaluations $B$, the overall regret approaches $0$ as $n$ and $B$ are increased.

# D  BASELINES

In this section, we first give a brief description of *Learning Initialization (LI)* (Wistuba et al., 2015) and *DistBO* (Law et al., 2019) used as baseline methods in the experiments. Then, we discuss the application of these methods to MSU-HPO.

## D.1  LEARNING INITIALIZATION

Learning Initialization (LI) by Wistuba et al. (2015) suggests promising hyperparameters by minimizing a meta-loss function. The meta-loss function is defined by the sum of a surrogated loss function on each source task. Intuitively, by minimizing the meta-loss function, LI can obtain hyperparameters that show good performance on average for source tasks.

## D.2  DISTBO

DistBO transfers knowledge across tasks using learnt representations of training datasets (Law et al., 2019). To measure similarity between these tasks, DistBO uses a distributional kernel, which learns the relationship between source and target tasks by joint Gaussian process model on hyperparameters and data representation. At the first iteration of optimization for the target task, DistBO uses LCB as the acquisition function to quickly select good hyperparameters[7].

DistBO models a joint distribution $p(x, y)$ of the training data for each task. However, in the MSU-HPO setting, modeling $p(x, y)$ for the target task is not possible because the labels on the target task is not available. Therefore, in our experiments, DistBO models the marginal distribution $p(x)$, not the joint distribution $p(x, y)$, for each task. This setting is also used in the original paper of DistBO (Section 5.1 in (Law et al., 2019)). If the covariate shift assumption holds, then from the optimization point of view, it is sufficient to model $p(x)$, not $p(x, y)$.

---

[7]Note that while our goal is to minimize the objective function, DistBO aims to maximize the objective function.

### D.3 DISCUSSION ON APPLICATION TO MSU-HPO

The major difference from the proposed method lies in the method of evaluating hyperparameters. While the proposed method evaluates one hyperparameter using all source tasks, LI and DistBO consider the situation where hyperparameters are evaluated for any one source task. The effect originated by this difference will become significant when the number of source tasks is large. For example, let us consider a scenario of optimization by DistBO where the available evaluation budget is $B = 100$. If the number of source tasks is 2, we can optimize each source task with evaluation budget $B = 50$ to assign a uniform evaluation budget to each source task. On the other hand, if the number of source tasks is 50, the evaluation budget for each source task is only $B = 2$, which makes optimization very difficult. Actually, in the Appendix C.5 in (Law et al., 2019), the existence of 1230 hyperparameter evaluations on source tasks is assumed on Parkinson's dataset. In contrast, our method works properly even if there is a limited evaluation budget for source tasks. In our experiment on Parkinson's dataset (in Section 4.2), we assume that only 50 hyperparameter evaluations on source tasks are available.

## E EXPERIMENTAL SETTING

### E.1 SETTINGS

For fair comparison, we used Gaussian Process Upper Confidence Bound (GP-UCB) (Srinivas et al., 2010) as a hyperparameter optimization algorithm for source tasks in all the optimization methods. For example of the proposed method, this corresponds to **OPT** in Algorithm 1. In our experiments, the confidence parameter in GP-UCB is fixed to 2 (following the setting used in (Nguyen et al., 2016; González et al., 2016; Bogunovic et al., 2018)).

We set the number of evaluations (i.e., $B$ in Algorithm 1 and 2) to 50. At the beginning of optimization, we sample 5 initial points uniform randomly. A Matérn $5/2$ kernel is used in the implementation of GP-UCB. Note that our study is formulated as a minimization, so we actually use LCB (Lower Confidence Bound) instead of UCB as an acquisition function. We used *densratio_py*[8] to estimate the density ratio by uLSIF (Kanamori et al., 2009). All the experiments were conducted on Google Cloud Platform (n1-standard-4) or MacBook Pro (2.2 GHz Intel Core i7, 16 GB).

For LI, which utilizes the gradient descent algorithm to optimize a meta-loss function defined by source tasks, we need to set a learning rate $\eta$ and a number of epochs $E$. Following (Wistuba et al., 2015), we set $\eta = 10^{-3}$ and $E = 10^3$. Note that we need one hyperparameter to be evaluated for the target task in MSU-HPO, so the number of hyperparameters we obtain by LI is one. To obtain the results of DistBO in our experiments, we use the authors' implementation[9]. We use a Matérn $5/2$ kernel for a fair comparison with other methods, whereas the default kernel in the code implementation is a Matérn $3/2$ kernel.

### E.2 HYPERPARAMETER OPTIMIZATION

Table 3 and 4 show the hyperparameter search spaces ($\Theta$) of SVM on the Parkinson's telemonitoring dataset and LightGBM on the GvHD dataset. We treat integer-valued hyperparameters as a continuous variable and rounded off before evaluations. For SVM, we used Radial Basis Function kernel in *scikit-learn* (Pedregosa et al., 2011). We used *microsoft/LightGBM*[10] to implement LightGBM. For the details of these parameters, please refer to the scikit-learn documentation[11] and the LightGBM documentation[12].

We normalized the feature vectors of the GvHD dataset as a preprocessing procedure. In contrast, in the Parkinson dataset, we did not apply standardization or normalization, because we have confirmed that these operations cause performance degradation. In the Parkinson dataset, we selected a task (patient) with the maximum number of data as the target task. In contrast, the tasks in the GvHD

---

[8]https://github.com/hoxo-m/densratio_py

[9]https://github.com/hcllaw/distBO

[10]https://github.com/microsoft/LightGBM

[11]https://scikit-learn.org/stable/modules/generated/sklearn.svm.SVR.html

[12]https://lightgbm.readthedocs.io/en/latest/Parameters.html

dataset all have the same number of data, thus we selected the task (patient) with the first task index as the target task. For statistical correctness, in the proposed unbiased and variance reduced estimators, we used different sources of data for the density ratio estimation and for the training of ML models. Specifically, we used $30\%$ of the training set for the density ratio estimation and the remaining $70\%$ for the learning of the ML models.

Table 3: Details of hyperparameters of SVM on Parkinson's telemonitoring dataset.

| Hyperparameters | Type | Scale | Search Space |
|---|---|---|---|
| Kernel Coefficient | float | log | $[5.0 \times 10^{-5}, 5.0 \times 10^{3}]$ |
| L2 Regularization Parameter | float | log | $[5.0 \times 10^{-5}, 5.0 \times 10^{3}]$ |

Table 4: Details of hyperparameters of LightGBM on GvHD dataset.

| Hyperparameters | Type | Scale | Search Space |
|---|---|---|---|
| Max Depth for Tree | int | linear | $[2, 6]$ |
| Feature Fraction | float | linear | $[0.1, 1.0]$ |
| Learning Rate | float | log | $[1.0 \times 10^{-3}, 1.0 \times 10^{-1}]$ |
| L2 Regularization Parameter | float | log | $[5.0 \times 10^{-5}, 5.0 \times 10^{3}]$ |

