# OpenReview forum: "Multi-Source Unsupervised Hyperparameter Optimization"
_ICLR.cc/2021/Conference — Reject_

### Official Review · AnonReviewer3 · 2020-10-21
**An interesting, well-written paper introducing a method of reweighting labelled pairs from multiple sources to optimize a machine learning model for an unlabelled dataset.**

**Rating:** 5
**Confidence:** 4

**Review:**

Strong points:
- Well-written and easy to read and follow.
- Results show the method does what it promises to.

Weaknesses:
- While it was easy to follow the paper's rationale, I found it difficult to motivate. Until the final experiment, I was left wondering what kind of application would have these constraints but also where the assumptions would be reasonable. This point I think should be easy to address given a small rework of the intro or perhaps a running example to periodically come back to.
- After reading about the last experiment, I found myself wondering why this solution is billed as a hyperparameter optimization solution; it sounds to me that the parameters of the model are also being optimized along the way. Again, this is a question of clarification and can easily be addressed.
- The fact that the Naive method beats the other two baselines and performs comparably to the unbiased proposed method makes me wonder whether (a) these are challenging enough tasks, or (b) those are competitive baselines.
- The density estimator and the divergence estimator are moving parts the errors of which perhaps warrant quantifying. For example, in both experiments the true labels were known and the authors can measure the error in the divergence estimate.

I'd be happy to increase my score if the above points are addressed.

---

> ### Author Response · Authors · 2020-11-16
> **Reply to AnonReviewer3**
>
> We would like to thank the reviewer for his/her useful, detailed feedback.
> We will update the paper with the suggested minor revisions and respond below to some concrete questions/comments.
>
> > While it was easy to follow the paper's rationale, I found it difficult to motivate.
>
> Our HPO setting, MSU-HPO, succeeds for the first time in the setting that there is no labeled data in the target task, by leveraging the data of multiple source tasks. This setting also appears in socially important social fields, such as medical healthcare [1,2,3,4]. Specifically, you can use our method when you have labeled training data from some (source) hospitals but you want to have a prediction model that works well on a target hospital different from the source hospitals. Our method does not depend on a specific ML model and can be widely applied. We will add such examples that motivate the importance and practicality of the MSU-HPO setting.
>
> [1] Qi Dou, Cheng Ouyang, Cheng Chen, Hao Chen, and Pheng-Ann Heng. Unsupervised cross-modality domain adaptation of convnets for biomedical image segmentations with adversarial loss.arXiv preprint arXiv:1804.10916, 2018.
> [2] Faisal Mahmood, Richard Chen, and Nicholas J Durr. Unsupervised reverse domain adaptation for synthetic medical images via adversarial training.IEEE transactions on medical imaging,37(12):2572–2581, 2018.
> [3] Christian S Perone, Pedro Ballester, Rodrigo C Barros, and Julien Cohen-Adad. Unsupervised domain adaptation for medical imaging segmentation with self-ensembling.NeuroImage,194:1–11, 2019.
> [4] Yifan Zhang, Ying Wei, Peilin Zhao, Shuaicheng Niu, Qingyao Wu, Mingkui Tan, and JunzhouHuang.  Collaborative unsupervised domain adaptation for medical image diagnosis.arXivpreprint arXiv:1911.07293, 2019.
>
>
> > After reading about the last experiment, I found myself wondering why this solution is billed as a hyperparameter optimization solution; it sounds to me that the parameters of the model are also being optimized along the way. Again, this is a question of clarification and can easily be addressed.
>
> Thank you for your comment. Yes, you are true that models are trained using training data with our variance reduced estimator as their objective during HPO. As we discussed in the Introduction part that there are some previous solutions for training predictors under the covariate shift setting. However, we argue that there is no work that explores how HPO should be done in similar settings.  Therefore, in this work, we focus on a new HPO setting we call MSU-HPO.
>
> In addition, the model training (statistical machine learning) and the hyperparameter optimization are based on completely different formulations (though they seem to be similar).  Therefore, even if there are some similar methods that are known to work well on the model training tasks, developing an HPO method for the corresponding problem setting is of independent interest.
>
>
> > The fact that the Naive method beats the other two baselines and performs comparably to the unbiased proposed method makes me wonder whether (a) these are challenging enough tasks, or (b) those are competitive baselines.
>
> The poor performances of DistBO and LI are not that strange, as they are the methods for conventional HPO problems, not for MSU-HPO. The HPO experiments are very challenging for the proposed method due to the small number of training data. If the number of data and the number of evaluations are large enough, the proposed method can find the optimal solution, as we analyzed in Appendix C. In contrast, DistBO and LI do not have such a guarantee.
>
> DistBO and LI are not methods for MSU-HPO and are not strong enough as baseline methods. However, as far as we know, there is no method that suits MSU-HPO. We think that MSU-HPO is most close to the DistBO setting, and we believe that the experimental comparisons are reasonable.

---

### Official Review · AnonReviewer4 · 2020-10-23
**An interesting papers on an importance sampling estimator applied to HPO with certain restrictions**

**Rating:** 6
**Confidence:** 3

**Review:**

**Summary**
In the situation where a given objective is computed with samples from a distribution, e.g. loss on validation data in hyperparameter optimization, this paper proposes a method to construct a surrogate objective using objectives computed on sets of samples each of which is from a different distribution. Basic idea is to use a linear combination of importance sampling estimators. Moreover, the optimal linear combination coefficients are identified in a sense of being optimal in a certain family of convex combination coefficients. This approach has an interesting application that hyperparameters of machine learning deployed on an unseen dataset, importantly, without labels(output) can be optimized as long as the distribution of the input of the unseen dataset is samplable.


**Strengths**
1. The paper provides an algorithm that can estimate an objective computed on data without requiring access to labels in some optimal sense.
2. Based on the proposed estimator, a transfer hyperparameter optimization algorithm to solve interesting problems is introduced.


**Weaknesses**
1. Maybe the novelty of the paper mainly lies in the combination not on inventing something new, which is, however, not certain since my coverage of relevant literature was not extensive.
2. I can imagine that someone may ask for more experiments of a large scale or of the type exemplified in the intro. On the other hand, the experiments can be regarded as designed concisely to demonstrate the authors' main points.


**Recommendation**
Overall, I am willing to defend the acceptance of this paper. This combination of transfer HPO and importance sampling estimator seem novel, interesting, and well-demonstrated, with which many interesting applications can be imagined.


**Questions**
- On the line right below eq.(2), the loss function L is assumed to be bounded. Is this condition is necessary in proofs of any theoretical ones? It seems that, in all experiments, all losses are unbounded.


**Additional feedback**
- Explicitly emphasizing that Def 3 is motivated by eq.(6) may guide readers better.
- While reading the paper, the questions arose were mostly answered after a few lines. The reading was pleasant for me and the paper is well-structured.

---

> ### Author Response · Authors · 2020-11-16
> **Reply to AnonReviewer4**
>
> We would like to thank the reviewer for his/her useful, detailed feedback.
> We will update the paper with the suggested minor revisions and respond below to some concrete questions/comments.
>
>
> > On the line right below eq.(2), the loss function L is assumed to be bounded. Is this condition is necessary in proofs of any theoretical ones?
>
> Yes, the bounded assumption is needed to ensure that the variance of the estimators does exist. We will clarify this point in the future revision.
>
>
> > Explicitly emphasizing that Def 3 is motivated by eq.(6) may guide readers better.
>
> Thank you for pointing it out. We will definitely emphasize it and clarify the relevant sentences.
>
>
> > While reading the paper, the questions arose were mostly answered after a few lines. The reading was pleasant for me and the paper is well-structured.
>
> Thank you for the positive comment!

---

### Official Review · AnonReviewer1 · 2020-10-24
**The paper proposes a theoretically-grounded solution for transfer learning in HPO without target labels. More baselines and challenging experiments would help.**

**Rating:** 6
**Confidence:** 4

**Review:**

The paper introduces multi-source unsupervised hyperparameter optimization (MSU-HPO), a novel BO framework where a range of related tasks are available but labels cannot be accessed for the target task. As ground truth on the target task is unavailable, the work introduces two estimators to approximate the target task objective. This enables HPO to be run to optimize the hyperparameters on the target task, converging faster to a good hyperparameter configuration.


Positive

1. **Significance.** To my knowledge, this is the first paper to investigate a transfer learning setting for HPO where labels are unavailable for the target task. This is of interest in several practical applications (e.g., advertising, as the authors discuss). The exploration of a new problem together with the introduction of principled estimators make both the paper's goal and methodology significant.
2. **Clarity.** The paper was a pleasure to read. Very clear and well structured. I am also confident about reproducibility as enough details about the algorithm and experimental setup are provided.
3. **Rigorous evaluation.** I appreciated the solid theoretical analysis paired with fully-controlled synthetic experiments where the degree of task similarity can be regulated and results compared against ground truth. All figures are based on multiple runs and have clearly detailed error bars.

Negative

1. **Easy experiments.** The experiments are run on synthetic data and on real-data with only two ML models (SVM and LightGBM). This is secondary considering that the theoretical analysis is solid, but tuning a wider range of ML algorithms (such as neural networks/NAS) would have made the case even stronger by showing that transfer learning is possible across a diverse class of models. The dimensionality of the optimized hyperparameter spaces is also very small, with respectively two and four tuned hyperparameters for SVM and LightGBM. Applying the method to more challenging scenarios would further demonstrate the benefits of the proposed approach.
2. **Not many baselines.** The main baselines the method is compared against are LI and DISTBO. While many transfer learning baselines are inapplicable as most assume target labels to be available (as discussed in the related work), this is not the case for all of them. An example is Feurer, et al.: Initializing Bayesian hyperparameter optimization via meta-learning, AAAI, 2015. This is not referenced but should be discussed, as it only uses hyperparameter configurations from previous related tasks to warm-start the new optimization. As this does not look at the labels of the target task, it could be compared against. This is also the case for Perrone et al., 2019, which learn a search space purely based on previous tasks and does not require target labels. Another search space pruning method that is not compared against nor discussed is Wistuba, et al.: Hyperparameter search space pruning–a new component for sequential model-based hyperparameter optimization, in Machine Learning and Knowledge Discovery in Databases, 2015.

Overall, I am inclined towards accepting this paper due to the rigorous theoretical evaluation, the significance of the problem, and the fairly satisfactory experimental evaluation (although more baselines and more challenging experiments would be needed to make a stronger case).

Additional questions

a. Related to the point above, how does the proposed approach compare to the transfer learning baselines references above that are purely based on learning an initialization or a good search space?

b. The naive method performs on part with LI in the synthetic experiments and clearly better than DISTBO in the real-world ones. Is this expected? Does this indicate that the chosen baselines (LI and DISTBO) may be too weak? In appendix D3 the authors discuss that this is probably due to the fact that only 50 hyperparameter evaluations on source tasks are available. Would it then not be useful to re-run the comparison under a varying number of hyperparameter evaluations available from source tasks? If the proposed method only outperforms baselines when a small number of prior evaluations are available, this should be clearly stated and shown.

c. Considering the relatively poor results of the unbiased estimator, would it not be better to re-frame the narrative to focus on the variance-reduced estimator? The unbiased one is an interesting ablation study, but given the results it might be better to clearly state from the beginning that the variance reduced estimator is the recommended choice.

d. Will code be made available?

Typos:
1. Page 12: "omited" ---> "omitted"
2. Figure 1b: "Comapring" ----> "Comparing"

---

> ### Author Response · Authors · 2020-11-16
> **Reply to AnonReviewer1**
>
> We would like to thank the reviewer for his/her useful, detailed feedback. We will update the paper with the suggested minor revisions and respond below to some concrete questions/comments.
>
>
> > a. Related to the point above, how does the proposed approach compare to the transfer learning baselines references above that are purely based on learning an initialization or a good search space?
>
> Thank you for the suggestion. The comparison with other methods with the search space pruning is interesting and insightful, and we would like to conduct this comparison in the future revision.
>
>
> > b. The naive method performs on part with LI in the synthetic experiments and clearly better than DISTBO in the real-world ones. Is this expected? Does this indicate that the chosen baselines (LI and DISTBO) may be too weak?
>
> These baseline methods (LI and DistBO) are, at least, powerful methods for a standard HPO setting with labeled target task, as reported in their papers. While we discuss the reason why their results were poor in Appendix D.3, no further details have been investigated yet.
>
> There is a possibility that HPO methods other than these methods will work well in MSU-HPO. Hence, a comprehensive comparison between other methods (pointed out by the reviewer) and the proposed methods is included in the future work.
>
>
> > Would it then not be useful to re-run the comparison under a varying number of hyperparameter evaluations available from source tasks?
>
> Thank you for the helpful comment. We agree that the suggested experiment is important in confirming the scope of application of the proposed method. We would like to note that the proposed method can unbiasedly estimate the target task objective in the MSU-HPO setting, but LI and DistBO do not. In other words, there is no guarantee that the estimation of LI and DistBO will match the true objective no matter how much the training data and the number of evaluations increase.
>
> Note that we analyze the dependence of the proposed method on the number of data and the number of evaluations of the estimation accuracy in Appendix C.
>
>
> > c. Considering the relatively poor results of the unbiased estimator, would it not be better to re-frame the narrative to focus on the variance-reduced estimator? The unbiased one is an interesting ablation study, but given the results it might be better to clearly state from the beginning that the variance reduced estimator is the recommended choice.
>
> Thank you for the helpful suggestion. We will try to improve the paper based on your comments.
>
>
> > d. Will code be made available?
>
> Yes, we have a plan to publicize our code upon publication.

---

> > ### Comment · AnonReviewer1 · 2020-11-23
> > **Thanks for the clarifications**
> >
> > I'd like to thank the authors for the clarifications. I hope the authors will discuss and compare to more baselines and I look forward to the paper revision.

---

### Official Review · AnonReviewer2 · 2020-10-30
**Interesting idea, but exposition often confusing, and results are lacking**

**Rating:** 3
**Confidence:** 3

**Review:**

The authors describe a method for training and tuning a machine learning model for a prediction task where no labels are available, and where thus no model can be fit in the standard supervised manner. Instead labels are estimated based on related tasks that do have labels. After this a predictor can be trained on those estimated labels, and can be tuned using a standard Bayesian optimization algorithm.

The main contribution consist in the description of an estimator for the labels. The paper also provides basic experimental results, both in the form of a naive toy example, and of results on two machine learning datasets.

### Questions / Comments
The paper is described (and titled) as a HP tuning paper. But to me it appears to mainly be concerned with unsupervised training, when there are related labeled datasets available, and should thus be described, analysed and tested mainly as such, and compared to other unsupervised learning techniques. The fact that it can be combined with any supervised learning method and can incorporate hyperparameter (HP) tuning is interesting, but the end results is still a unsupervised prediction model.

In this framing, the comparison only HP tuning algorithms not made for the specific setting does not seem to be the right comparison. Thus I argue that the results are not sufficient to show that the technique proposed can be useful in practice.

The results are compared to existing HP Tuning warm start algorithms by getting a single suggested HP configuration. One detail that I do not understand from the text is how the HPs found by the baseline are evaluated, if no available labels are assumed. Can the authors shed more light on this aspect?

As the authors themselves note, the baselines are not well suited for the setting, where we have a very low transfer budget to be split among many source tasks. I would expect much simpler baselines to perform much better, for example: 1) just using the default HPs of the given algorithm 2) running on a single arbitrary source task the whole budget as a simple BO task, and use the best found HPs of this source task. (But it is still not clear how finding HPs is useful if we have no labels, so I might be missing something major here).

The description of the baseline they call "Naive" is also not very clear. If would be good to have more details on this baseline.

In Experimental Procedure, What does (1) do? How is the ML model tuned if not by MSU-HPO, which seems to be done later. It is not clear to me from the text.

To summarize, while the method is interesting, it is insufficiently motivated, either as a special type of unsupervised learning, or if it is something different a stronger motivation of why this is worthwhile (you can plug in arbitrary models, use arbitrary HP, tuning algorithms or other reasons). Additionally, the experiments would benefit from clearer descriptions and stronger baselines.

### Typos
Figure 1b:
Comapring -> Comparing

Table 1 caption:
performs almost the same with naive in Parkinson given
their standard errors -> grammar should be improved

---

> ### Author Response · Authors · 2020-11-16
> **Reply to AnonReviewer2**
>
> We would like to thank the reviewer for his/her useful, detailed feedback.
> We will update the paper with the suggested minor revisions and respond below to some concrete questions/comments.
>
> > The paper is described (and titled) as a HP tuning paper. But to me it appears to mainly be concerned with unsupervised training, when there are related labeled datasets available
>
> First of all, thank you for making a significant and interesting point. Let us discuss this point in detail.
>
> We think that our work lies on HPO because the aim of this paper is to obtain the optimal set of hyperparameters, and the proposed method is general purpose for HPO (, i.e., the proposed method does not depend on any specific machine learning models). However, we completely agree that this work is different from conventional HPO approaches, which formulate HPO as black-box optimization. This means that, even if there are related datasets, the conventional HPO approaches do not use the training data (X, y) *explicitly* and perform optimization using only the information of the hyperparameters and their evaluation values. MSU-HPO cannot be solved without utilizing the information of these training data.
> As far as we know, for the first time in this study, we put something structure in the related data to streamline HPO explicitly, and perhaps this is what makes this study less like HPO. We agree that our approach is very unique in HPO; however, we strongly believe that this study will narrow the gap between the fields of HPO and statistical machine learning, where it is common to make explicit assumptions about data distributions. We hope that our work opens up new research directions of HPO.
>
>
> > One detail that I do not understand from the text is how the HPs found by the baseline are evaluated, if no available labels are assumed.
> > In Experimental Procedure, What does (1) do?
>
> The baseline methods such as “Naive” and “LI” perform optimization by using only source tasks, instead of an unlabeled target task. The best hyperparameter found in the optimization of source tasks is used to calculate the final evaluation value.
> This is described in Step (1) of ‘Experimental Procedure’ in Section 4.2. DistBO uses the unlabeled dataset of the target task as well as the labeled datasets of the source tasks (please see Appendix D.2 for the details).
>
>
> > As the authors themselves note, the baselines are not well suited for the setting
>
> Thank you for pointing it out. We agree that the baseline methods are not suitable for the MSU-HPO setting, as described in Appendix D.3. We appreciate the suggestion of baseline methods, and will compare that baselines in the future revision.
>
>
> > The description of the baseline they call "Naive" is also not very clear. If would be good to have more details on this baseline.
>
> In the MSU-HPO setting, the labeled data of the target task is not available for conventional HPO methods. Therefore, the “Naive” method uses the data of the labeled source tasks for training, and does not use any information about the unlabeled target task. Specifically, the “Naive” method prepares the concatenation of (labeled) source tasks, and splits it into training and validation data. After that, perform the experiment according to the procedure in Experimental Procedure of Section 4.2.

---

> > ### Comment · AnonReviewer2 · 2020-11-16
> > **The goal of the paper and baselines**
> >
> > I thank the authors for their answer. However I want to challenge the following point
> >
> > "the aim of this paper is to obtain the optimal set of hyperparameters"
> >
> > How HPO is generally motivated is: in practice there are many situations where model X or Y is used, but the HPs are suboptimal, or tuning them manually takes a lot of time or effort. So the general aim for supervised ML is to produce a good model according to some metric, and one way to improve many of the best models (the hyperparameters) is by HPO.
> >
> > We should not forget the overall goal (providing a good model), when looking at the subgoal (finding good HPs to further improve a model). The final goal in the task described in the paper appears to be to provide a regression model on unlabeled data, where labeled transfer learning data is not available, but labeled source tasks are. I don't understand why it is not simply framed as thus, and then (if it is the case) motivate why HPO is important in that setting.
> >
> > Once the problem is framed as such, better baselines (that do not need to use HPO) can come to mind. For example, just train a model for each source task, and predict their average or median, on the target task. A slight variation: train a single model where the source task is one hot encoded as a feature; then at prediction again one can use the mean or median of what would be the prediction for the various source tasks. An even simpler idea: concatenate all source tasks and just train a model on these, use the obtained model directly for predicting on the target. (Or is this Naive? My understanding from the paper is that only the HP configuration found this way is used, and then a new model is trained only using the target dataset and f_hat as an objective, but I found the section "Experimental Procedure" hard to follow). In each of these cases there is no need to perform HPO, just either use the default parameters or perform HPO on the source tasks.

---

> > > ### Author Response · Authors · 2020-11-20
> > > **Reply to AnonReviewer2**
> > >
> > > Thank you for pointing out the important points. We agree that the proposed method should be compared with such baselines, which are not limited to the HPO methods.
> > > However, we would like to note that such comparisons can lead to wrong conclusions in confirming the performance of the hyperparameters obtained.
> > > Let us suppose that we consider the following two baseline methods:
> > >
> > > MethodA: one source task is used to train an ML model.
> > > MethodB: all source tasks are used to train an ML model.
> > >
> > > In this case, even if the same hyperparameters are used, the performance of MethodA is different from that of MethodB. For example, when the distribution of source and target task is the same, the performance of MethodB is better than that of MethodA, even if the same hyperparameters are used.
> > > Therefore, this is problematic as a way to evaluate the quality of the selected hyperparameters.
> > > In contrast, since the same data are used when training an ML model in our experimental setting,  we can avoid such drawing incorrect conclusions.

---

### Decision · Program_Chairs · 2021-01-07
**Final Decision**

**Decision:**

Reject

**Comment:**

The paper has been actively discussed in the light of the authors’ response. Even though the paper was, overall, found quite clear with a solid theoretical support, the reviewers listed several concerns that remained unsolved after the rebuttal, e.g.,

* The proposed approach may not be properly scoped/positioned and evaluated as an HPO method, a concern unanimously shared across the reviewers. Although this is not a concern impossible to overcome, the reviewers believed it could not be achieved as part of a simple revision of the paper.
* The lack of challenging baselines to fully assess the performance of the proposed method (e.g., see the list suggested by Reviewer 1)
* Along the lines of the previous point, the experiments focus on small-scale settings, which does not make it possible to completely assess the performance of the approach
* Some discrepancy between the theoretical analysis and the actual experimental settings (e.g., the assumption about bounded losses not valid with the squared loss)

As illustrated by its scores, the paper is extremely borderline. Given the mixed perspectives of pros and cons, the paper is eventually recommended for rejection.
This list, together with the detailed comments of the reviewers, highlight opportunities to improve the manuscript for a future resubmission.